# SweRank: Software Issue Localization with Code Ranking

**Revanth Gangi Reddy**[*1,2] **Tarun Suresh**[*1] **JaeHyeok Doo**[*3] **Ye Liu**[2] **Xuan Phi Nguyen**[2]
**Yingbo Zhou**[2] **Semih Yavuz**[2] **Caiming Xiong**[2] **Heng Ji**[1] **Shafiq Joty**[2]
[1] University of Illinois at Urbana-Champaign   [2] Salesforce Research   [3] KAIST AI

## Abstract

Software issue localization, the task of identifying the precise code locations (files, classes, or functions) relevant to a natural language issue description (e.g., bug report, feature request), is a critical yet time-consuming aspect of software development. While recent LLM-based agentic approaches demonstrate promise, they often incur significant latency and cost due to complex multi-step reasoning and relying on closed-source LLMs. Alternatively, traditional code ranking models, typically optimized for query-to-code or code-to-code retrieval, struggle with the verbose and failure-descriptive nature of issue localization queries. To bridge this gap, we introduce SweRank[1], an efficient and effective retrieve-and-rerank framework for software issue localization. To facilitate training, we construct SweLoc, a large-scale dataset curated from public GitHub repositories, featuring real-world issue descriptions paired with corresponding code modifications. Empirical results on SWE-Bench-Lite and LocBench show that SweRank achieves state-of-the-art performance, outperforming both prior ranking models and costly agent-based systems using closed-source LLMs like Claude-3.5. Further, we demonstrate SweLoc's utility in enhancing various existing retriever and reranker models for issue localization, establishing the dataset as a valuable resource for the community.

## 1 Introduction

The scale and complexity of modern software systems continue to grow exponentially, with a significant portion of development effort dedicated to identifying and resolving software issues. This has fueled growth in automated software issue fixing (Cognition AI, 2024), with recent LLM-based patch generation (Yang et al., 2024a; Gauthier, 2024) solving real-world issues on benchmarks such as SWE-Bench (Jimenez et al., 2023), and commercial copilots integrating "one-click" quick-fix suggestions directly into IDEs (Microsoft, 2023; Cursor, 2025; Windsurf, 2025). Central to the process of fixing software issues is the task of **issue localization**: accurately identifying *where* in the codebase the necessary changes should be made. This involves pinpointing the specific files, classes, or functions relevant to a given issue description, typically provided in natural language (e.g., a bug report). Effective localization is critical; without correctly identifying the relevant code segments, any subsequent attempt at automated repair is likely to fail or, worse, introduce new faults.

Given the importance of localization, recent work treats it as an agentic reasoning problem (Yao et al., 2023) and has investigated the use of sophisticated LLM-based agents (Yang et al., 2024b; Yu et al., 2025; Chen et al., 2025) that issue commands such as 'read-file', 'grep' and 'traverse-graph' to iteratively explore codebases, navigate file structures, search for code patterns, and analyze dependencies. While powerful, these agent-based compound systems often involve multiple rounds of interaction ($\approx$7–10 on average) with large models and complex reasoning processes, which can incur considerable API costs ($\approx$\$0.66 per example with Claude-3.5) at high latency. Moreover, agent traces are brittle: they rely on temperature sampling and require complex tool orchestration.

An alternative, more efficient strategy is to frame issue localization as an information retrieval problem, specifically using code ranking models (Yue et al., 2021; Zhang et al., 2024; Suresh

---

*Equal Contribution. Work done during Revanth's internship at Salesforce AI Research.

[1]Code and models are available here: https://github.com/SalesforceAIResearch/SweRank

et al., 2024). Such models can directly rank candidate code snippets (e.g., functions or files) based on their relevance to a given natural language query, and quickly score and sort potential locations within a large codebase. However, prior code ranking models are still inferior in performance as they have predominantly been optimized for tasks distinct from issue localization. These typically include query-to-code retrieval (Li et al., 2024a), which aims to find code implementing a described functionality, and code-to-code retrieval (Wang et al., 2023a; Li et al., 2024b), focused on identifying semantically similar code fragments. The task of issue localization presents unique characteristics; input queries (issue descriptions) are often substantially more verbose than typical NL-to-code queries[2] and, more crucially, issues tend to describe observed erroneous behavior or system failures rather than specifying desired functionality. This fundamental difference in query nature and intent suggests that models trained on conventional code retrieval data (Husain et al., 2019; Suresh et al., 2024) may not be optimally suited for issue localization.

To bridge this gap, we introduce SWERANK, a code ranking framework trained specifically for software issue localization. SWERANK employs a standard yet effective retrieve-and-rerank architecture, comprising two core components: (1) SWERANKEMBED, a bi-encoder embedding model serving as the code retriever; and (2) SWERANKLLM, an instruction-tuned LLM serving as a code reranker. To train SWERANK, we construct SWELOC, a new large-scale issue localization dataset curated from public Github repositories, providing realistic training examples. SWERANKEMBED is trained using a contrastive objective, where the issue descriptions serve as queries, the known localized functions act as positive examples, and carefully mined code snippets from the same repository function as hard negatives. Subsequently, SWERANKLLM is trained as a list-wise reranker (Reddy et al., 2024); it takes as input the issue description alongside the top-$K$ candidates retrieved by SWERANKEMBED and predicts an improved ranking permutation, thereby enhancing the final localization.

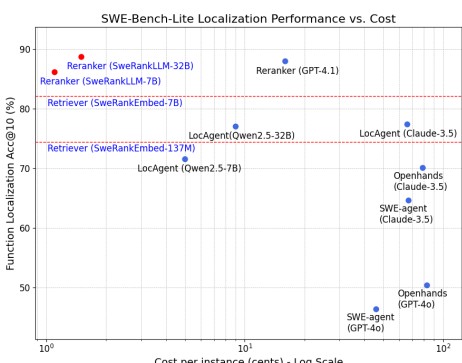

Figure 1: Comparison of localization performance versus cost per instance on SWE-Bench-Lite. Our proposed SWERANKEMBED retriever and SWERANKLLM reranker models achieve superior accuracy at a significantly lower cost compared to agent-based localization methods.

Empirical results demonstrate that SWERANK achieves state-of-the-art performance for file, module and function-level localization on Swe-Bench-Lite (Jimenez et al., 2023) and LocBench (Chen et al., 2025). Further, we show that SWERANK, built on open-source models, has a considerably better performance to cost ratio compared to agent-based approaches that employ closed-source LLMs like Claude-3.5 (Anthropic, 2023), as illustrated in Figure 1. Finally, we demonstrate the effectiveness of our SWELOC data by showing that it consistently improves localization performance when used for finetuning a variety of text and code-pretrained retriever and reranker models.

## 2 RELATED WORK

### 2.1 SOFTWARE ISSUE LOCALIZATION

Software issue localization or Fault Localization (FL) aims to identify the specific code locations responsible for reported bugs. Traditional fault localization methods (Wong et al., 2016) can be grouped into spectrum-based and program-analysis approaches. Spectrum-based fault localization (SFL) (de Souza et al., 2016; Amario de Souza et al., 2024) statistically associates test outcomes with executed code elements to rank statements or functions by their 'suspiciousness' based on passing and failing test coverage. Complementary static and dynamic analyses exploit program structure–through call-graph traversal (Adhiselvam et al., 2015), dependency analysis (Elsaka, 2017), or program slicing (Soremekun et al., 2021)–to constrain the search space of potential bug locations. While these methods provide a statistical basis for finding faults, they require precise program models and cannot leverage the rich natural language context in bug reports.

---

[2] 460 tokens in SWE-Bench (Jimenez et al., 2023) issues vs 12 tokens in CSN (Li et al., 2024a) queries.

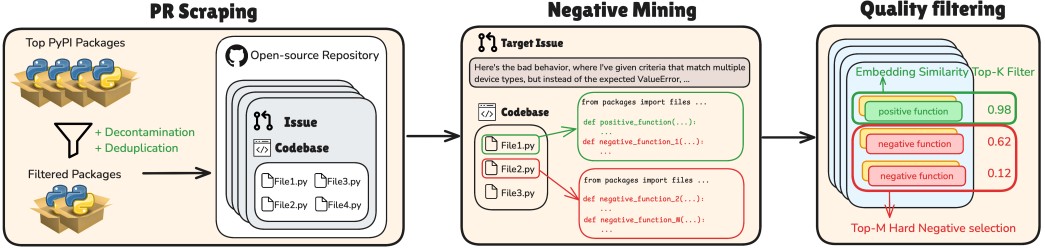

Figure 2: Overview of SWELOC data construction pipeline, illustrating the three main stages.

Modern approaches instead use LLM-based agent frameworks that treat bug localization as a planning and searching problem. AgentFL (Qin et al., 2024) incorporates a multi-agent system with a three step procedure involving interpreting the bug context, traversing the codebase and verifying the suspected fault. OpenHands (Wang et al., 2025) and SWE-Agent (Yang et al., 2024b) use bash commands or custom interfaces to navigate repositories and access files. Other agentic systems combine IR with tool use: MoatlessTools (Örwall, 2024) integrates a semantic code search engine into an agent's loop to guide it to relevant files. More recently, LocAgent (Chen et al., 2025) constructs a graph of the codebase for an LLM agent to do multi-hop reasoning over code dependencies. While these agent-driven approaches have achieved impressive results, they incur substantial costs and have high latency. Agent-based methods must orchestrate multiple steps of reasoning and tool use, which makes them brittle; a single failure in the chain (e.g., a misleading intermediate query or an incomplete code observation) can derail the entire localization process. SWERANK instead formulates issue localization as a single-shot ranking problem, which is highly efficient and cost-effective.

## 2.2 CODE RANKING

Transformer-based code ranking models (Wang et al., 2023c; Zhang et al., 2024; Günther et al., 2023; Suresh et al., 2024) have set state-of-the-art on a variety of code retrieval tasks (Li et al., 2024a;b) by learning joint embeddings of text and code. Wang et al. (2023c) and Zhang et al. (2024) learn improved code representations by incorporating a mix of training objectives, such as span denoising, text-code matching and causal LM pretraining, over large-scale code corpora such as CodeSearchNet (Husain et al., 2019) and The Stack (Kocetkov et al., 2022). Suresh et al. (2024) improve the contrastive training process between function snippets and associated doc-strings with better consistency filtering and harder negative mining. Liu et al. (2024b) incorporate multi-task contrastive data that includes code contest generation (Billah et al., 2024), code summarization (Sontakke et al., 2022), code completion (Liu et al., 2024a), code translation (Pan et al., 2024) and code agent conversation (Jin et al., 2024). However, prior code ranking models rarely include error logs in their training data and are not optimized for issue localization, where queries are verbose bug reports rather than precise functionality requests. In contrast, SWERANK is explicitly trained on SWELOC, a new automatically collected set of real-world issue reports paired with known buggy functions. By optimizing a bi-encoder retriever and a listwise LLM reranker on this task-specific data, SWERANK directly aligns verbose bug descriptions with faulty code, thereby improving localization accuracy.

## 3 SWELOC: ISSUE LOCALIZATION DATA

Existing code retrieval datasets (Husain et al., 2019; Suresh et al., 2024) are generally valuable for tasks like NL-to-code search which mainly requires functionality matching. However, they are sub-optimal for training models aimed at software issue localization. The nature of software issues–often detailed descriptions of failures rather than concise functional specifications–necessitates a dataset that accurately reflects this challenge of precisely identifying the problematic functions. To address this gap and provide a suitable training ground for our SWERANK framework, we constructed SWELOC, a novel large-scale dataset specifically curated for the task of localizing code snippets relevant to software issues. SWELOC is derived from real-world software development activities captured in public GitHub repositories. Our methodology comprises three main phases: (1) identifying and filtering relevant pull requests (PRs) from popular Python repositories (§3.1), (2) processing these

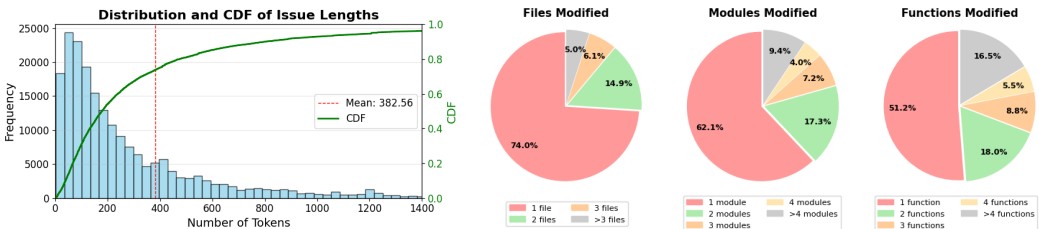

Figure 3: (Left) Distribution of query lengths in the SweLoc dataset. The red dashed line indicates a mean query length of 382.56 tokens, underscoring the detailed nature typical of issue reports. (Right) Distribution of the number of (a) files, (b) modules, and (c) functions modified per GitHub issue. This highlights that while many localizations are concentrated, a significant number span multiple code units, particularly at finer granularities.

PRs to extract issue descriptions paired with their corresponding code modifications (§3.2), and (3) applying consistency filtering and hard-negative mining to enhance the quality of training instances (§3.3). An overview of this process is shown in Figure 2.

## 3.1 IDENTIFYING RELEVANT PRS

Our data collection involves selecting the repositories associated with the top 11,000 PyPI packages on GitHub. To ensure repository quality and relevance to our task, we apply several filtering criteria. Repositories are required to contain at least 80% Python code. To prevent data leakage and overlap with existing benchmarks, we exclude repositories already present in SWE-Bench (Jimenez et al., 2023) and LocBench (Chen et al., 2025). Finally, we perform deduplication based on source code overlap to remove near-identical repositories. This process results in a curated set of 3387 repositories.

Following the SWE-Bench methodology, we identify pull requests (PRs) within these repositories that (1) resolve a linked GitHub issue and (2) include modifications to test files, indicating the issue resolution was verified. For each such PR, we collect the issue description and the codebase snapshot at the PR's base commit. This procedure results in 67,341 initial (PR, codebase) pairs. Figure 3 provides further details on the dataset's composition, including query and repository edit distributions.

## 3.2 LOCALIZATION PROCESSING

Using the collected (PR, codebase) pairs, we create contrastive training data in the form of ⟨query, positive, negatives⟩ tuples. For each tuple, the issue description serves as the query. Each function modified within the PR is designated as a positive example, corresponding to a distinct training instance. Thus, a PR modifying $N$ functions yields $N$ training instances. The negatives for each instance come from the unmodified functions within the corresponding codebase. This initial set of instances are further refined via consistency filtering and hard-negative mining, as described next.

## 3.3 CONSISTENCY FILTERING AND HARD NEGATIVES

The quality of ⟨query, positive, negatives⟩ tuples used for training significantly impacts the ranking model performance (Suresh et al., 2024). Effective contrastive learning requires relevant positives and challenging negatives (semantically similar to the positive but irrelevant to the query). However, issue descriptions in open-source repositories can be vague, leading to noisy signals for relevance between the issue descriptions and associated code modifications when directly used for training.

To mitigate this, we employ filtering and mining techniques following recent work (Günther et al., 2023; Suresh et al., 2024). First, we apply top-$K$ consistency filtering (Suresh et al., 2024) to retain only instances where the positive code snippet is semantically close to the query relative to other code snippets in the repository. Formally, given an instance $i$ with issue description $t_i$, a positive function $c_i$, and the set of other unrelated functions $F_i$ in the repository, we use a pre-trained embedding model (CODERANKEMBED (Suresh et al., 2024)) to compute similarities between $t_i, c_i$ and all functions in $F_i$. Instance $i$ is retained only if $c_i$ ranks within the top $K$ functions in $F_i$, based on similarity to $t_i$. We set $K = 20$, with ablation studies in §5.3.1.

Beyond filtering for relevance of positive pairs, incorporating challenging negatives is crucial for enabling the model to distinguish between semantically similar instances (Moreira et al., 2024). To this end, we employ a hard negative mining strategy that leverages the previously computed similarities to select a set of hard negatives $B_i = \{c_j^-\}_{j=1}^M$ for each instance $i$. These negatives $c_j^-$ are chosen from $F_i$ such that they are among the top $M$ (=15) most similar functions to the query $t_i$.

# 4  SWERANK METHODOLOGY

In this section, we present our proposed ranking framework for software issue localization. SWERANK adopts a two-stage retrieve-and-rerank approach with two key components: (1) SWERANKEMBED, a bi-encoder retriever that efficiently narrows down candidate code snippets from large codebases; and (2) SWERANKLLM, a listwise LLM reranker that refines these initial results for improved localization accuracy. Next, we elaborate on the architecture and training objectives for these components.

## 4.1  SWERANKEMBED

The retriever component, SWERANKEMBED, utilizes a bi-encoder architecture (Reimers & Gurevych, 2019) to generate dense vector representations for GitHub issues and code functions within a shared embedding space. Let $(t_i, c_i^+)$ represent a positive pair from the SWELOC dataset, consisting of an issue $t_i$ and the corresponding code function modified $c_i^+$. The bi-encoder maps these to embeddings $(h_i, h_i^+)$, derived from the last hidden layer of the encoder. For a training batch of size $N$, let $H = \{h_i^+\}_{i=1}^N$ denote the set of positive code embeddings. Let $H_B = \bigcup_{i=1}^n \{h_{ij}^-\}_{j=1}^M$ be the set of embeddings for the $M$ hard negatives mined for each issue $t_i$ in the batch (as described in §3.3).

SWERANKEMBED is trained using an InfoNCE contrastive loss (Oord et al., 2018). The objective encourages the embedding $h_i$ of an issue to have a higher similarity with its corresponding positive code embedding $h_i^+$, compared to its similarity with all other $h_k^+$ embeddings ($k \neq i$) and all hard negative embeddings $h_{kj}^-$ within the batch. The loss for a single positive pair $(h_i, h_i^+)$ is:

$$\mathcal{L}_{\mathcal{CL}} = -\log \left( \frac{\exp(\mathbf{h}_i \cdot \mathbf{h}^+{}_i)}{\sum_{\mathbf{h}_k \in (\mathbf{H_B} \cup \mathbf{H})} \exp(\mathbf{h}_i \cdot \mathbf{h}_k)} \right) \tag{1}$$

The denominator sums over the positive embedding $h_i^+$ itself and $N(M+1)-1$ negative embeddings relative to $h_i$. During inference, candidate code functions for a given issue description are ranked based on the cosine similarity between their respective embeddings and the issue embedding.

## 4.2  SWERANKLLM

For the reranking stage, we employ SWERANKLLM, an instruction-tuned LLM for reranking. SWERANKLLM adopts a listwise ranking approach (Pradeep et al., 2023b), which offers better performance than pointwise methods by considering the relative relevance of candidates. Typically, listwise LLM rerankers are trained to process an input consisting of the query and a set of candidate documents, each associated with a unique identifier. The model's training objective is then to generate the full sequence of identifiers, ordered from most to least relevant according to the ground-truth ranking. However, since SWELOC does not provide a ground-truth ranking among the negative functions for the issue $t_i$, generating a complete target permutation for training is not feasible.

To adapt listwise reranking training to our setting where only the positive is known, we modify the training objective. Formally, let $\mathcal{D} := \{d_i\}_{i=1}^{|\mathcal{D}|}$ be a training dataset of triplets, where each sample $d_i := (t_i, c_i^+, \{c_{i,j}^-\}_{j=1}^M)$ includes a GitHub issue $t_i$, a relevant positive code $c_i^+$, and a set of $M$ irrelevant negative codes $\{c_{i,j}^-\}_{j=1}^M$. We first assign a unique numerical identifier from 1 to $M+1$) to each function in the set $c_i^+ \cup \{c_{i,j}^-\}_{j=1}^M$. Let $I_i^+$ be the identifier assigned to the positive function $c_i^+$. Instead of training the model to predict the full ranked list of identifiers, we train it to correctly generate the identifier corresponding to the single positive function, $I_i^+$. Thereby, the training objective for a given sample $d_i$ is thus simplified to maximizing the likelihood of the first generated (i.e. top-ranked) identifier:

$$\mathcal{L}_{LM} = -\log(P_\theta(I_i^+|x)) \tag{2}$$

| Type | Method | Model | File (%) | | | Module (%) | | Function (%) | |
|------|--------|-------|---------|---------|---------|---------|---------|---------|---------|
| | | | Acc@1 | Acc@3 | Acc@5 | Acc@5 | Acc@10 | Acc@5 | Acc@10 |
| Agent | MoatlessTools Örwall (2024) | GPT-4o | 73.36 | 84.31 | 85.04 | 74.82 | 76.28 | 57.30 | 59.49 |
| | | Claude-3.5 | 72.63 | 85.77 | 86.13 | 76.28 | 76.28 | 64.60 | 64.96 |
| | SWE-agent Yang et al. (2024b) | GPT-4o | 57.30 | 64.96 | 68.98 | 58.03 | 58.03 | 45.99 | 46.35 |
| | | Claude-3.5 | 77.37 | 87.23 | 90.15 | 77.74 | 78.10 | 64.23 | 64.60 |
| | Openhands Wang et al. (2025) | GPT-4o | 60.95 | 71.90 | 73.72 | 62.41 | 63.87 | 49.64 | 50.36 |
| | | Claude-3.5 | 76.28 | 89.78 | 90.15 | 83.21 | 83.58 | 68.25 | 70.07 |
| | LocAgent Chen et al. (2025) | Qwen2.5-7B(ft) | 70.80 | 84.67 | 88.32 | 81.02 | 82.85 | 64.23 | 71.53 |
| | | Qwen2.5-32B(ft) | 75.91 | 90.51 | 92.70 | 85.77 | 87.23 | 71.90 | 77.01 |
| | | Claude-3.5 | 77.74 | 91.97 | 94.16 | 86.50 | 87.59 | 73.36 | 77.37 |
| Retriever | BM25 (Robertson et al., 1994) | | 38.69 | 51.82 | 61.68 | 45.26 | 52.92 | 31.75 | 36.86 |
| | Jina-Code-v2 (161M) (Günther et al., 2023) | | 43.43 | 71.17 | 80.29 | 63.50 | 72.63 | 42.34 | 52.19 |
| | Codesage-large-v2 (1.3B) (Zhang et al., 2024) | | 47.81 | 69.34 | 78.10 | 60.58 | 69.71 | 33.94 | 44.53 |
| | CodeRankEmbed (137M) (Suresh et al., 2024) | | 52.55 | 77.74 | 84.67 | 71.90 | 78.83 | 51.82 | 58.76 |
| | SFR-Embedding-2 (7B) (Meng et al., 2024) | | 58.03 | 80.29 | 83.94 | 70.07 | 79.20 | 56.20 | 64.23 |
| | GTE-Qwen2-7B-Instruct (7B) (Li et al., 2023) | | 65.33 | 82.85 | 89.78 | 76.28 | 83.58 | 63.14 | 70.44 |
| | SWERANKEMBED-SMALL (137M) (Ours) | | 66.42 | 86.50 | 90.88 | 79.56 | 85.04 | 63.14 | 74.45 |
| | SWERANKEMBED-LARGE (7B) (Ours) | | 72.63 | 91.24 | 94.16 | 84.31 | 89.78 | 71.90 | 82.12 |
| + Reranker | CodeRankLLM (7B) (Suresh et al., 2024) | | 72.99 | 89.78 | 93.80 | 85.04 | 90.88 | 71.90 | 83.58 |
| | GPT-4.1 | | 82.12 | 95.62 | 97.08 | 93.07 | 93.43 | 81.75 | 87.96 |
| | SWERANKLLM-SMALL (7B) (Ours) | | 78.10 | 92.34 | 94.53 | 89.05 | 92.70 | 79.56 | 86.13 |
| | SWERANKLLM-LARGE (32B) (Ours) | | 83.21 | 94.89 | 95.99 | 90.88 | 93.43 | 81.39 | 88.69 |

Table 1: Performance (in %) on SWE-Bench-Lite. The rerankers use SWERANKEMBED-LARGE as the retriever. Gray corresponds to results with closed-source models. Best retriever numbers are in blue, while best overall numbers (except GPT-4.1) are in **bold**.

where $x$ is the input prompt constructed from the issue $t_i$ and the set of candidate functions $c_i^+ \cup \{c_{i,j}^-\}_{j=1}^M$ along with their assigned identifiers, and $P_\theta$ represents the listwise LLM reranker.

During training, we omit the end-of-sequence token after predicting $I_i^+$ to retain the model's capability to generate full ranked lists for inference, as required by the listwise format. As we show later in our experiments in §5.3.2, our approach enables finetuning any listwise reranker for the software issue localization task, without needing the full candidate ranking ordering for training supervision.

# 5 EXPERIMENTS

The experiments compare SWERANK's performance against state-of-the-art agent-based localization methods, and other code ranking models (§5.2). Furthermore, we investigate the impact of our SWELOC dataset, analyzing how its quality controls (such as consistency filtering) and size influence model performance (§5.3.1), and examining its generalizability by evaluating effectiveness in finetuning various pre-existing retriever and reranker models for the issue localization task (§5.3.2).

## 5.1 SETUP

**Model Training:** We train the SWERANK models in two sizes: *small* and *large*. All models are finetuned using our SWELOC dataset. SWERANKEMBED-SMALL is initialized with CodeRankEmbed (Suresh et al., 2024), a SOTA 137M code embedding model, while the large variant is initialized with GTE-Qwen2-7B-Instruct (Li et al., 2023), a 7B parameter text embedding model employing Qwen2-7B-Instruct as its encoder. The small version of SWERANKLLM is initialized with CODERANKLLM (Suresh et al., 2024), a 7B parameter code-pretrained listwise reranker. The large version is initialized with Qwen-2.5-32B-Instruct that is pretrained using text listwise reranking data (Pradeep et al., 2023b). More details in Appendix A.

**Baselines:** Our primary comparison is against prior agent-based localization methods. Specifically, we include OpenHands (Wang et al., 2025), SWE-Agent (Yang et al., 2024b), MoatlessTools (Örwall, 2024) and LocAgent (Chen et al., 2025), the current SOTA agent-based approach. Notably, these methods predominantly use closed-source models, with LocAgent also finetuning open-source models

for this task. For the retrieve-and-rerank framework, we compare SWERANKEMBED-SMALL against BM25 (Robertson et al., 1994) and several code embedding models of comparable size, including Jina-Code-v2 (Günther et al., 2023), Codesage-large-v2 (Zhang et al., 2024), and CodeRankEmbed (Suresh et al., 2024). For the 7B parameter embedding model comparison, we include GTE-Qwen2-7B-Instruct, which ranks third on the MTEB leaderboard (Muennighoff et al., 2023) at the time of evaluation. For the reranker comparison, we include CODERANKLLM and other closed source-models such as GPT-4.1. Due to the larger size of LocBench, comparisons on this benchmark are limited to a subset of the best-performing baselines.

**Datasets & Metrics:** We evaluate on SWE-Bench-Lite (Jimenez et al., 2023) and LocBench (Chen et al., 2025). Following Suresh et al. (2024), we exclude examples from SWE-Bench-Lite where no existing functions were modified by the patch, resulting in 274 retained examples out of 300. While SWE-Bench-Lite primarily consists of bug reports and feature requests, LocBench ( 560 examples) also includes security and performance issues. Consistent with Chen et al. (2025), we measure localization performance at three granularities: file, module (class) and function, with Accuracy at $k$ (Acc@k) as the evaluation metric. This metric deems localization successful if all relevant code locations are correctly identified within the top-$k$ results. The relevance score for a specific file or module is determined by the maximum score of any function contained within that file or module.

## 5.2 LOCALIZATION RESULTS

Table 1 compares performance of different localization methods on the SWE-Bench-Lite benchmark. The results indicate that our SWERANK models surpasses the performance of all evaluated agent-based methods. Furthermore, the SWERANKEMBED-SMALL model, despite its relatively small size of 137M parameters, demonstrates highly competitive performance, outperforming prior 7B parameter embedding models. Notably, SWERANKEMBED-LARGE achieves higher Acc@10 for function localization than LocAgent with Claude-3.5. Employing the SWERANKLLM reranker subsequently enhances the retriever's output, establishing a new SOTA for localization performance on this benchmark across all granularities. Qualitative examples are provided in Appendix H.

Table 2 shows results on LocBench. A similar trend is observed, with the large variants of SWERANKEMBED and SWERANKLLM setting new SOTA performance. Figure 4 provides a detailed breakdown of localization accuracy across the four distinct difficulty categories within LocBench. Despite being primarily trained with bug reports in SWELOC, the SWERANK models demonstrate impressive generalizability across other categories.

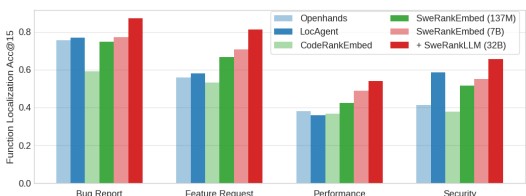

Figure 4: Localization performance across different categories within LocBench. SWERANK considerably outperforms Agent-based methods using Claude-3.5.

## 5.3 ANALYSIS

Our analysis presented in this section aims to demonstrate the following key points: 1) the impact of SWELOC data quality and size on final model performance (§5.3.1); 2) the utility of SWELOC for finetuning various retriever and reranker models (§5.3.2; and 3) the cost-effectiveness of the proposed SWERANK framework (§5.3.3). Unless otherwise mentioned, the results are on SWE-Bench-Lite.

### 5.3.1 DATA QUALITY AND SIZE

Public GitHub repositories, as a source for contrastive data, often contain noisy instances. This study first examines the effectiveness of consistency filtering (§3.3), specifically the influence of the positive-rank threshold, $K$. This parameter dictates the minimum rank of the instance's positive (relative to negatives, based on similarity with the issue description) for inclusion of the instance in the training set. Increasing $K$ relaxes the filtering, yielding more training instances but potentially introducing more noise. As shown in Figure 5a, finetuning SWERANKEMBED-SMALL with SWELOC data filtered by different $K$ values reveals that optimal performance is achieved with a moderate $K$ (e.g., $K$=20), striking a balance between instance quality and dataset size. The absence of filtering ($K$=None) proves detrimental as performance drops after finetuning compared to pre-trained model.

| Method | Loc Model | File (%) | | Module (%) | | Function (%) | |
|--------|-----------|------|------|------|------|------|------|
| | | Acc@5 | Acc@10 | Acc@10 | Acc@15 | Acc@10 | Acc@15 |
| Agentless | `Claude-3.5` | 67.50 | 67.50 | 53.39 | 53.39 | 42.68 | 42.68 |
| OpenHands | `Claude-3.5` | 79.82 | 80.00 | 68.93 | 69.11 | 59.11 | 59.29 |
| SWE-agent | `Claude-3.5` | 77.68 | 77.68 | 63.57 | 63.75 | 51.96 | 51.96 |
| LocAgent | `Qwen2.5-7B(ft)` | 78.57 | 79.64 | 63.04 | 63.04 | 51.43 | 51.79 |
| | `Claude-3.5` | 83.39 | 86.07 | 70.89 | 71.07 | 59.29 | 60.71 |
| Retriever | CodeRankEmbed (137M) | 74.29 | 80.36 | 63.93 | 67.86 | 47.86 | 50.89 |
| | GTE-Qwen2-7B-Instruct (7B) | 75.54 | 82.50 | 67.14 | 71.61 | 51.79 | 57.14 |
| | SWERANKEMBED-SMALL (137M) | 80.36 | 84.82 | 71.43 | 75.00 | 58.57 | 63.39 |
| | SWERANKEMBED-LARGE (7B) | 82.14 | 86.96 | 75.54 | 78.93 | 63.21 | 67.32 |
| + Reranker | CodeRankLLM (7B) | 83.93 | 88.21 | 76.96 | 80.89 | 64.64 | 69.29 |
| | GPT-4.1 | 85.89 | 88.75 | 79.64 | 82.50 | 71.61 | 74.64 |
| | SWERANKLLM-SMALL (7B) | 85.54 | 88.39 | 79.11 | 82.14 | 69.46 | 74.46 |
| | SWERANKLLM-LARGE (32B) | **86.61** | **89.82** | **81.07** | **83.21** | **71.25** | **76.25** |

Table 2: Performance (in %) on LocBench. The rerankers use SWERANKEMBED-LARGE as the retriever. Gray correspond to results with closed-source models. Best retriever model numbers are in blue, while best overall numbers (except GPT-4.1) are in **bold**.

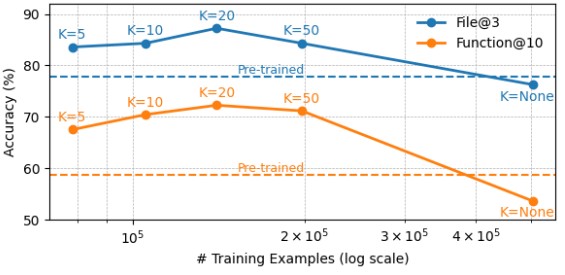
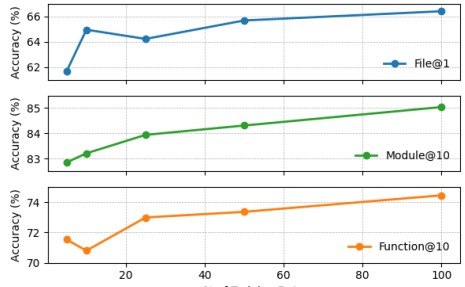

(a) While accuracy improves from doing consistency filtering, i.e. discarding instances where the positive's rank among negatives is $>K$, no filtering ($K$=*None*) hurts performance.

(b) All metrics show a general upward trend as the percentage of training data ($K$=20) increases.

Figure 5: Impact of (a) training data filtering and (b) data size on SWERANKEMBED-SMALL performance.

Controlling for data quality (by fixing $K$=20), the impact of dataset size is investigated. Figure 5b illustrates that training with varying proportions of the filtered data yields considerable performance improvements, even with only 5% of the data. Generally, larger dataset sizes correspond to further performance gains. These experiments underscore the significance of both data quality and quantity, demonstrating that merely increasing data volume without quality control can be detrimental. Appendix §E further examines the impact of negative hardness on model performance.

### 5.3.2 CHOICE OF RETRIEVER AND RERANKER

Here, we demonstrate the effectiveness of SWELOC by showing improvements for a variety of retriever and reranker models from finetuning. First, the following embedding models, pre-trained on different data types, are finetuned for one epoch on SWELOC: Arctic-Embed (Merrick et al., 2024), primarily pre-trained on English text retrieval data; CodeRankEmbed, pre-trained on 22 million NL-to-Code examples (Suresh et al., 2024); and Arctic-Embed-v2.0 (Yu et al., 2024),

| Base Retriever | Pretrain | Func. Acc@10 (%) |
|----------------|----------|------------------|
| CodeRankEmbed | English+Code | 59.5→**72.3** (+12.8) |
| Arctic-Embed | English | 53.7→71.9 (+17.4) |
| Arctic-Embed-v2.0 | Multilingual | 62.0→70.1 (+8.1) |

Table 3: Accuracy (Before→After) from finetuning different retrievers with SWELOC data.

pre-trained on a mix of English and multilingual data. From Table 3, we see all models showing significant performance improvement from finetuning. Notably, models that initially performed weaker (e.g., Arctic-Embed) showed greater gains. This outcome validates that SWELOC can substantially improve the performance of *any* embedding model for software issue localization.

Next, text- and code-instruction-tuned LLMs of different sizes from the Qwen2.5 family (Yang et al., 2024c; Hui et al., 2024) are finetuned as listwise LLM rerankers using SWELOC data. Since we only apply loss on the first generation token, to ensure compatibility with the listwise output format, all models were initially pretrained on listwise text reranking data (Pradeep et al., 2023b), which provides the full ranking order to use for supervision. The results, shown in Table 4, indicate that rerankers across different model sizes universally benefit from finetuning on SWELOC. An interesting observation is that the code-pretrained model performs marginally better at the 7B scale, while the text-pretrained models achieve better results at the 3B and 32B scales. Results with finetuning Llama-3.1 are in Appendix B.

| Base LLM Reranker | Func. Acc@5 (%) | Func. Acc@10 (%) |
|---|---|---|
| Qwen-2.5-Text (32B) | 77.0→**81.4** (+4.4) | 82.5 →**86.1** (+3.6) |
| Qwen-2.5-Code (32B) | 76.3→79.9 (+3.6) | 81.8 →84.7 (+2.9) |
| Qwen-2.5-Text (7B) | 75.2→75.6 (+0.4) | 81.4 →82.5 (+1.1) |
| Qwen-2.5-Code (7B) | 75.5→75.9 (+0.4) | 81.0 →83.6 (+2.6) |
| Qwen-2.5-Text (3B) | 68.3→73.7 (+4.6) | 76.6→82.5 (+5.9) |
| Qwen-2.5-Code (3B) | 71.2→71.9 (+0.7) | 80.3→81.0 (+0.7) |

Table 4: Localization accuracy (Before→After) from finetuning different listwise rerankers with SWELOC.

### 5.3.3 INFERENCE COST ANALYSIS

Agent-based localization approaches typically involve multiple iterations, each requiring extensive chain-of-thought generation (Wang et al., 2023b), incurring considerable cost at inference. In contrast, SWERANK offers significant cost-effectiveness as the SWERANKLLM reranker only needs to generate output candidate identifiers to determine the ranking order. Furthermore, the SWERANKEMBED output embeddings can be pre-computed, resulting in negligible extra cost. Table 5 compares the inference costs of SWERANKLLM with other agent-based methods. Clearly, agent-based approaches, often relying on closed-source models for better performance, are highly cost-intensive. SWERANK is substantially cheaper while providing significantly better performance, with up to *6X* better performance-cost tradeoffs compared to LocAgent.

| Method | Model | Cost($) ↓ | $\frac{\text{Acc@10}}{\text{Cost}}$ ↑ |
|---|---|---|---|
| SWE-agent | GPT-4o | 0.46 | 0.8 |
| | Claude-3.5 | 0.67 | 1.0 |
| Openhands | GPT-4o | 0.83 | 0.6 |
| | Claude-3.5 | 0.79 | 0.9 |
| LocAgent | Claude-3.5 | 0.66 | 1.2 |
| | Qwen2.5-7B(ft) | 0.05 | 13.2 |
| | Qwen2.5-32B(ft) | 0.09 | 8.6 |
| Reranker | GPT-4.1 | 0.16 | 5.9 |
| | SWERANKLLM (7B) | 0.011 | **79.0** |
| | SWERANKLLM (32B) | 0.015 | 57.5 |

Table 5: SWERANKLLM has considerably better inference cost-efficiency than agent-based methods while being more performant.

### 5.3.4 IMPACT ON DOWNSTREAM ISSUE RESOLUTION

This section analyzes the impact of improved localization on downstream code repair performance. To evaluate issue resolution, we utilize SWE-Fixer (Xie et al., 2025), a two-step pipeline consisting of code file retrieval (localization) followed by code editing. We compare the repair outcomes on SWE-Bench-Lite when employing different localization methods: the native localization mechanism of SWE-Fixer, LocAgent (with Claude-3.5), our SWERANK (large variant), and an oracle. The oracle simulates perfect localization by using the ground-truth edited file, thereby providing an upper bound for the repair framework. From Table 6, we see that

| Localization | File Acc@1 | Repair Pass@1 |
|---|---|---|
| SWE-Fixer | 69.7 | 21.0 |
| LocAgent | 78.5 | 22.6 |
| SWERank | **83.2** | **24.5** |
| Oracle | 100 | 25.9 |

Table 6: Impact of localization accuracy on downstream issue resolution.

better localization provided by SWERANK yields improved issue resolution, with oracle results showing that repair performance is currently constrained by the code editing model.

### 5.3.5 PERFORMANCE ANALYSIS BY ISSUE COMPLEXITY

Aggregate metrics often obscure performance variance on complex issues. To address this, we stratify the test set by num_gold (the number of functions modified in the ground truth patch) as a proxy for issue complexity. We compare our approach against LocAgent and Gemini-Embedding, with results summarized in Figure 6.

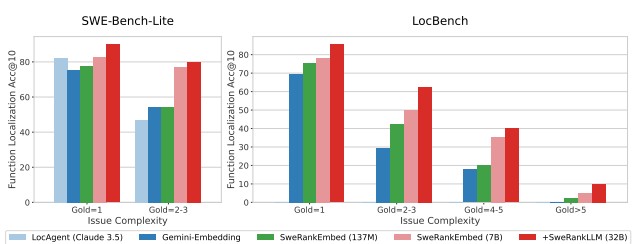

Figure 6: Function Acc@10 breakdown by issue complexity.

As expected, performance degrades as the number of modified functions increases. Instances requiring changes to $> 5$ functions are extremely difficult for all models, resulting in single-digit accuracy. However, SWERANKEMBED-LARGE demonstrates significantly better scaling on complex issues (num_gold=2–3) compared to Agentic approaches; on SWE-Bench-Lite, LocAgent drops from 82.0% to 47.1%, while our retriever maintains 77.1%. Furthermore, the reranker consistently improves performance across all complexity levels, confirming that it successfully captures cross-function dependencies that the bi-encoder might miss.

### 5.3.6 PERFORMANCE ANALYSIS BY LEXICAL AND SEMANTIC OVERLAP

To further dissect the model's capabilities, we analyzed performance by grouping instances based on **Lexical Overlap** (Rouge-1) and **Semantic Overlap** (Cosine Similarity).

**Lexical Overlap.** We bucket instances into four groups using the Rouge-1 score between the issue description and the ground-truth localized functions. A high Rouge score indicates significant keyword overlap. Figure 7 summarizes the results. We observe that performance generally degrades as lexical overlap decreases. However, even in the

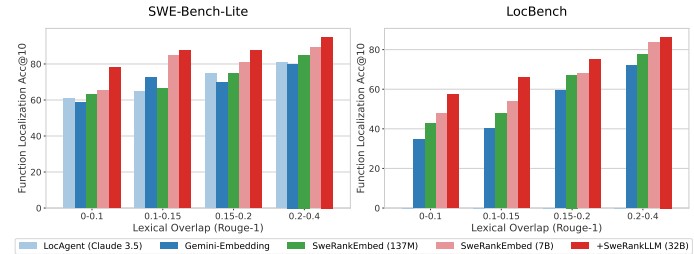

Figure 7: Performance breakdown by **Lexical Overlap** (Rouge-1).

lowest overlap bucket (0.0–0.1), SWERANKEMBED-LARGE outperforms LocAgent (65.2% vs 60.9% on SWE-Bench-Lite), demonstrating that our model does not rely solely on keyword matching. Furthermore, SWERANKLLM consistently improves performance, with significant gains seen specifically for instances with low lexical overlap.

**Semantic Overlap.** We also categorize instances based on the mean cosine similarity (computed via Gemini-Embedding) of the issue description and the ground-truth functions. As shown in Figure 8, performance is directly correlated with semantic overlap, achieving near-perfect accuracy for high-similarity instances ($> 0.8$). No-

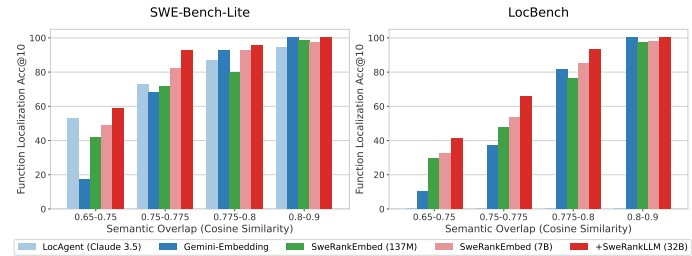

Figure 8: Performance breakdown by **Semantic Overlap**.

tably, SWERANKLLM reranker considerably boosts performance over the retriever in low-similarity buckets (0.65–0.75), outperforming the multi-turn LocAgent approach. This suggests that while agentic tool-use can bridge the semantic gap, our training on SWELOC—which incorporates hard negatives—enables the SWERANK framework to learn these non-obvious mappings effectively without the cost of agentic inference.

## 6 CONCLUSION

This paper frames software issue localization as a specialized ranking task and introduces SWERANK, a highly performant and cost-effective retrieve-and-rerank framework. To effectively train SWERANK models, we construct SWELOC, a large-scale contrastive training dataset derived from real-world GitHub issues, employing consistency filtering and hard-negative mining for quality. Empirical evaluations on SWE-Bench-Lite and LocBench demonstrate state-of-the-art localization performance using SWERANK, significantly outperforming costly closed-source agent-based systems. The introduction of SWELOC dataset provides a valuable resource for advancing research in this domain.

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
