# A    TRAINING DETAILS

## A.1    SWERANKEMBED

Our data filtering, negative mining, and model finetuning are implemented using the contrastors package (Nussbaum et al., 2024). The SWERANKEMBED-SMALL encoder uses CODERANKEMBED, which was initialized with Arctic-Embed-M (Merrick et al., 2024), a text encoder supporting an extended context length of 8,192 tokens and pretrained on large-scale web query- document pairs, along with public text retrieval datasets (Yang et al., 2018; Kwiatkowski et al., 2019; Thorne et al., 2018). The encoder supports a query prefix "*Represent this query for searching relevant code:* ", as set by (Suresh et al., 2024). The model is finetuned using 8 GH200 GPUs for two epochs with a learning rate of 2e-5, a batch size of 64 and 15 hard negatives per example.

The SWERANKEMBED-LARGE encoder uses GTE-Qwen2-7B-Instruct (Li et al., 2023), which was pretrained on a large corpora of text retrieval data. For this model, we use a custom query prefix "*Instruct: Given a github issue, identify the code that needs to be changed to fix the issue. Query:* ". The model is finetuned using 8 GH200 GPUs for 1 epoch with a learning rate of 8e-6, a batch size of 64 and 7 hard negatives per example.

## A.2    SWERANKLLM

**Training data:**    For each <query, positive, negatives> tuple from SWELOC, we randomly sample 9 negative codes to fit the listwise reranking window size of 10 along with the positive code. To prevent the positional bias from affecting the reranker and ensure model robustness (Pradeep et al., 2023a), we shuffle the order of candidate codes for each training example. Since the combined length of a GitHub issue and corresponding candidate codes may exceed the model's maximum embedding size, we set the maximum length per candidate code to 1024 and the total length limit to 16348. For overlong prompts, we truncate the query to reach the maximum total length. This preserves meaningful context for issue localization as much as possible within the limited context window size for effective model training. The rerankers are all first pretrained with text listwise reranking data (Pradeep et al., 2023b) to teach the model to follow the listwise output format.

**Hyperparameters:**    For the LLM reranker training, with both text reranking and SWELOC data, we trained for one epoch with a global batch size of 128, an initial learning rate of 5e-6 with 50 warmup steps, cosine learning rate scheduler, bfloat16 precision, and noisy embeddings (Jain et al., 2023) with a noise scale $\alpha = 5$. For efficient long-context, multi-gpu training, we used DeepSpeed (Rasley et al., 2020) ZeRO stage 3 with 16 GH200 GPUs.

# B    EXPERIMENTS WITH MORE RERANKER MODELS

To demonstrate the broader applicability of our dataset, we conduct experiments with finetuning Llama-3.1 8B Instruct (Grattafiori et al., 2024) as a listwise reranker. The models are first pre-trained on general text reranking data from RankZephyr (Pradeep et al., 2023b) and subsequently finetuned on our SWELOC dataset. Results, shown in Table 7, demonstrate significant performance gains on both SWE-Bench-Lite and LocBench after fine-tuning on SWE-

| Method Type | SWE-Bench-Lite | | LocBench | |
|---|---|---|---|---|
| | Acc@5 | Acc@10 | Acc@5 | Acc@10 |
| Zeroshot Reranker | 60.22 | 81.39 | 61.96 | 69.11 |
| RankZephyr finetune | 72.99 | 80.29 | 64.11 | 70.00 |
| + SWELOC finetune | **77.01** | **85.77** | **68.04** | **73.04** |

Table 7: Function localization accuracy of Llama-3.1 8B Instruct as a listwise LLM reranker.

Loc. This confirms that our dataset is a valuable resource for improving the issue localization capabilities of various LLM families, not just Qwen 2.5.

# C    RETRIEVER CEILING ANALYSIS

To assess the upper bound (performance ceiling) provided by the retrieval stage, we report extended metrics (Acc@20, @50, and @100) in Table 8. We compare our models against Gemini-Embedding (Lee et al., 2025). On SWE-Bench-Lite, SWERANKEMBED-LARGE achieves a retrieval

| Model | SWE-Bench-Lite (Acc@$K$) | | | | LocBench (Acc@$K$) | | | |
|---|---|---|---|---|---|---|---|---|
| | Acc@10 | Acc@20 | Acc@50 | Acc@100 | Acc@10 | Acc@20 | Acc@50 | Acc@100 |
| SweRankEmbed-Small | 74.45 | 81.75 | 87.96 | 91.97 | 58.57 | 67.50 | 75.71 | 82.32 |
| SweRankEmbed-Large | **82.12** | **86.50** | **90.88** | **93.43** | **63.21** | **71.25** | **80.71** | **84.29** |
| Gemini-Embedding | 72.26 | 79.20 | 87.96 | 90.88 | 51.43 | 60.18 | 70.00 | 78.39 |

Table 8: Extended retrieval metrics. Acc@$K$ indicates the percentage of instances where all ground-truth functions are within the top-$K$ retrieved candidates. The Acc@100 score indicates a performance ceiling for the subsequent reranker.

| Model | Depth | Width | Dim | Acc@5 | Acc@10 |
|---|---|---|---|---|---|
| SweRankEmbed-Small (137M) | 12 | 768 | 768 | $51.82 \rightarrow 63.14$ | $58.76 \rightarrow 74.45$ |
| Qwen3-Embedding-0.6B | 28 | 2048 | 1024 | $52.55 \rightarrow 66.79$ | $62.77 \rightarrow 75.18$ |
| SweRankEmbed-Large (7B) | 28 | 3072 | 3584 | $63.14 \rightarrow 71.90$ | $70.44 \rightarrow 82.12$ |
| Qwen3-Embedding-8B | 36 | 3072 | 4096 | $60.95 \rightarrow \mathbf{73.72}$ | $71.53 \rightarrow \mathbf{83.94}$ |

Table 9: Impact of retriever model capacity. We compare different variants of the SWERANKEMBED and Qwen3-Embedding models. Performance is reported as (Before → After) finetuning on SWELOC.

ceiling (Acc@100) of 93.43%. Given that SWERANKLLM-LARGE achieves 88.7% Acc@10, the gap suggests that the retrieval stage is not the primary bottleneck. Furthermore, this retrieval ceiling is significantly higher than the best performance achieved by agentic methods like LocAgent ($\sim$78%).

# D   ABLATION STUDIES ON MODEL CAPACITY

While the experimental results in the main text demonstrates that performance generally improves with model size for rerankers, we provide additional experiments here to analyze the sensitivity of the retriever to model capacity and architecture design.

**Impact of Encoder Depth & Width:**   To isolate the effects of model architecture, we compare our SWERANKEMBED variants against the recently released Qwen3-Embedding models (Zhang et al., 2025) (0.6B and 8B variants). We finetuned the Qwen3 models on the SWELOC dataset using the exact same procedure as SWERANKEMBED. Table 9 details the model specifications and performance. Comparing SWERANKEMBED-SMALL to Qwen3-0.6B, we observe moderate gains from utilizing a significantly deeper and wider encoder. However, comparing SWERANKEMBED-LARGE to Qwen3-8B suggests diminishing returns from further increasing depth (28 vs. 36 layers). Conversely, the considerable performance gap between Qwen3-0.6B and SWERANKEMBED-LARGE appears driven by the larger embedding dimension, which we investigate below.

**Impact of Embedding Dimension:**   To strictly isolate the impact of embedding dimension, we performed a controlled ablation using the Qwen3-Embedding-0.6B model, which supports flexible vector dimensions via Matryoshka Representation Learning (MRL) (Kusupati et al., 2022). Table 10 presents the results on SWE-Bench-Lite after finetuning with MRL.

| Dimension ($D$) | Acc@5 (Before → After) | Acc@10 (Before → After) |
|---|---|---|
| 1024 | $52.55 \rightarrow \mathbf{66.79}$ | $62.77 \rightarrow \mathbf{75.18}$ |
| 512 | $50.00 \rightarrow 65.69$ | $59.85 \rightarrow 74.09$ |
| 256 | $44.16 \rightarrow 59.12$ | $52.92 \rightarrow 69.71$ |
| 128 | $39.05 \rightarrow 56.93$ | $46.72 \rightarrow 66.79$ |
| 64 | $33.21 \rightarrow 50.00$ | $38.32 \rightarrow 60.22$ |

Table 10: Controlled ablation on embedding dimension using Qwen3-Embedding-0.6B with Matryoshka Representation Learning. SWELOC provides larger relative gains at lower dimensions.

Performance drops significantly as embedding size decreases, identifying dimension as a critical factor. Interestingly, finetuning on SWELOC yields larger relative gains at lower dimensions (e.g., +21.9 points Acc@10 for $D = 64$ vs. +12.5 points for $D = 1024$), highlighting the dataset's utility even for compressed representations.

## E    BENEFIT OF ITERATIVE NEGATIVE MINING

Here, we demonstrate the benefit of iterative negative mining by examining the impact of negative hardness on SWERANKEMBED performance. Figure 9 shows localization accuracy for Large and Small variants (finetuned and pretrained) with increasingly hard negatives. In an iterative mining approach, 1st iteration negatives are mined using the small pretrained model, and 2nd iteration negatives use the small model from 1st iteration. Results indicate that finetuning with random negatives yields smaller gains, while using 2nd iteration negatives yields notably improves performance over the 1st iteration.

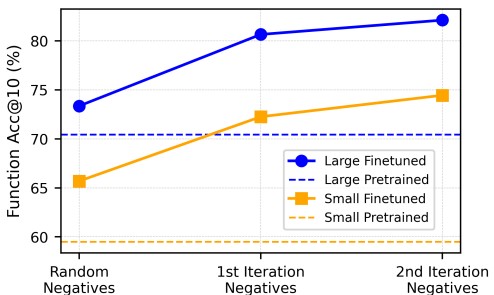

Figure 9: Finetuned models notably improve from an additional iteration of negative mining.

## F    MULTILINGUAL GENERALIZATION

Although SWELOC is constructed primarily from Python repositories, we hypothesize that SWERANK generalizes effectively to other languages because the underlying base models (CODERANKEMBED and GTE-QWEN2) are pretrained on massive multilingual corpora.

To empirically validate this, we evaluate the models on SWE-Bench Multilingual (Yang et al., 2025), which includes 234 tasks across 9 languages (C, C++, Java, JavaScript, TypeScript, Rust, PHP, Ruby, and Go). We compare the performance against Gemini-Embedding (Lee et al., 2025), a state-of-the-art proprietary general-purpose retriever.

| Method | Model | Acc@5 | Acc@10 |
|---|---|---|---|
| | CodeRankEmbed (137M) | 26.50 | 35.04 |
| | SweRankEmbed-Small (137M) | 33.33 | 44.02 |
| Retriever | GTE-Qwen2-7B-Instruct (7B) | 34.19 | 42.31 |
| | SweRankEmbed-Large (7B) | **39.74** | **50.85** |
| | Gemini-Embedding | 36.75 | 47.44 |
| | SweRankEmbed-Small (Base) | 33.33 | 44.02 |
| Reranker | + CodeRankLLM | 42.74 | 51.28 |
| | + SweRankLLM-Small | **49.15** | **56.84** |

Table 11: Function Localization Performance on SWE-Bench Multilingual. SWERANK generalizes effectively to non-Python languages.

The results, summarized in Table 11, support our hypothesis. SWERANKEMBED-LARGE (50.85% Acc@10) outperforms the proprietary Gemini-Embedding (47.44%), despite being finetuned on Python data. Furthermore, finetuning on SWELOC provided significant gains over the base models for both the retriever and the reranker. This demonstrates that the "issue-to-code" relevance signal learned from SWELOC is not language-specific and transfers effectively across different languages.

## G  EFFICIENCY COMPARISON WITH AGENTIC BASELINE

### G.1  INFERENCE LATENCY ANALYSIS

To complement the cost analysis, we evaluate the inference latency of our approach compared to LocAgent. The average latency is measured over 50 instances on SWE-Bench-Lite. For the SWERANK framework, the retrieval embeddings can be pre-computed and indexed, making the online retrieval cost negligible; therefore, the primary latency bottleneck stems purely from the reranking step.

| Approach | Model | Latency (s) |
|---|---|---|
| SweRank | SweRankLLM (7B) | **12.5** |
| SweRank | GPT-4o | 30.2 |
| LocAgent | GPT-4o | 85.3 |

Table 12: Inference Latency comparison.

Table 12 summarizes the results. When deploying the SWERANKLLM-SMALL model locally on a single 80GB A100 GPU, the system achieves an average latency of just 12.5 seconds per instance. This is approximately **7×** **faster** than the LocAgent baseline, making SWERANK far more viable for real-time developer assistance scenarios where rapid feedback is critical. Even when controlling for the underlying model by using GPT-4o for both approaches, SWERANK remains nearly **3×** **faster**. This efficiency gain primarily comes from SWERANK resolving the issue in a single-turn ranking pass, whereas agentic baselines like LocAgent relying on multi-turn loops, involving iterative thought generation, tool execution, and context reading, which naturally accumulates significant latency.

### G.2  TOKEN EFFICIENCY AND FOOTPRINT

Beyond latency, the computational load of a system is heavily influenced by its token usage. Agent-based approaches often suffer from "context bloat," as they must maintain a running history of all past thoughts, observations, and tool outputs throughout the interaction loop. We analyzed the average token footprint (Average Input Prompt & Output Tokens) required for each github issue.

| Approach | Prompt Tokens | Output Tokens |
|---|---|---|
| SweRank | 78,409 | 741 |
| LocAgent | 234,197 | 1,884 |

Table 13: Token footprint comparison.

As shown in Table 13, SWERANK operates with a $\sim$**3×** **lower** input token footprint compared to the agentic baseline. By formulating localization as a ranking problem rather than a sequential decision-making process, SWERANK eliminates the need for extensive history management. Furthermore, the reduction in output tokens is even more pronounced. Since output tokens are significantly more expensive and slower to generate than input tokens, this reduction directly translates to the lower latency observed in §G.1 and substantially reduced inference costs. This confirms that SWERANK provides a more sustainable and scalable alternative to agentic loops for issue localization.

## H  QUALITATIVE EXAMPLES

Figure 10 presents qualitative examples from SWE-Bench-Lite where SWERANK correctly localizes the function to edit while LocAgent is unable to. In both instances, LocAgent incorrectly identifies functions that likely correspond to where the problem manifests rather than where it originates.

## I  DIVERSITY OF ISSUE TOPICS IN SWELOC

To provide more insight into the variety and complexity of issue topics in SWELOC, we analyze the distribution of topics for 10k randomly sampled instances. We use Nomic Atlas[3], a popular unstructured text visualization tool, that employs a cluster-based keyword identification algorithm and leverages a language model to generate topics. Figure 11 shows the frequency of top-15 topics.

---

[3]https://atlas.nomic.ai/

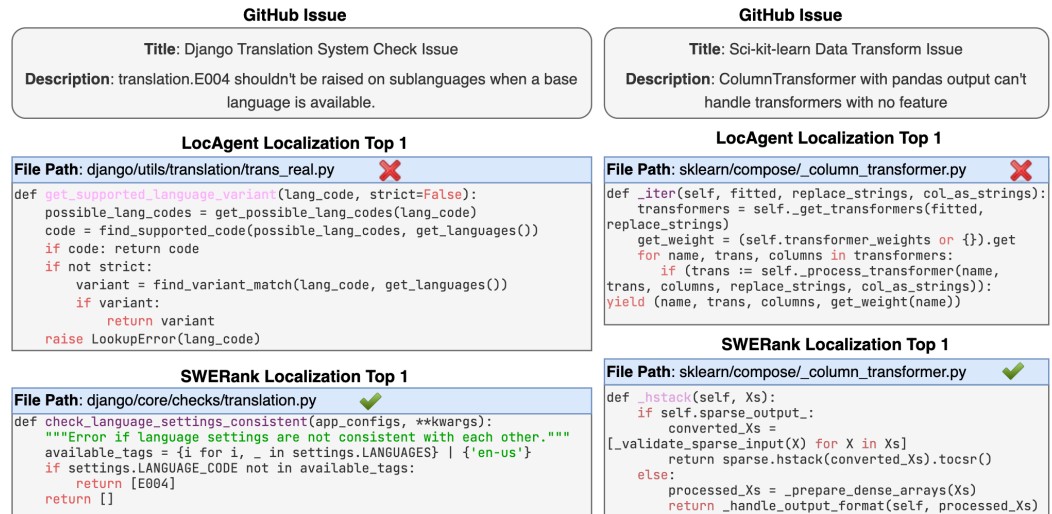

Figure 10: Examples from SWE-Bench-Lite where LocAgent mislocalizes the function, while our SWERank framework does function localization correctly

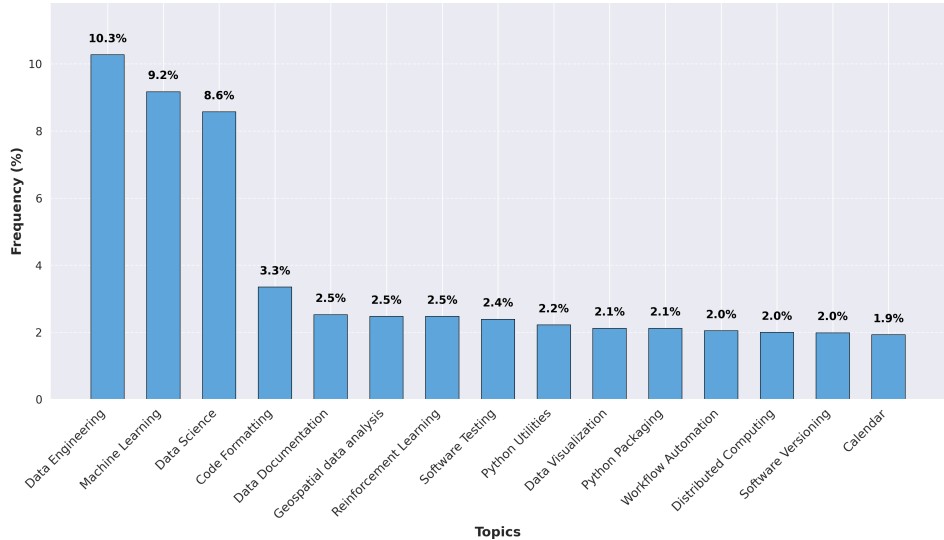

Figure 11: Top-15 issue topics and their frequencies from a randomly sampled subset of SWELOC.