# OpenReview forum: "SWERank: Software Issue Localization with Code Ranking"
_ICLR.cc/2026/Conference — ICLR 2026 Poster_

### Official Review · Reviewer_ok1L · 2025-10-20

**Soundness:** 3
**Presentation:** 3
**Contribution:** 3
**Rating:** 6
**Confidence:** 3

**Summary:**

The paper proposes SWERANK, a retrieve-and-rerank framework for software issue localization. It combines  (1) SWERANKEMBED, a
bi-encoder embedding model serving as the code retriever; and (2) SWERANKLLM, an instruction-tuned
LLM serving as a code reranker. To support training, the authors curate SWELOC, a large dataset pairing real GitHub issue descriptions with the functions/files modified in the corresponding fixes. On SWE-Bench-Lite and LocBench, SWERANK achieves new SOTA across file/module/function granularity while being far cheaper than multi-step agent systems using closed LLMs; ablations analyze data quality filters (consistency threshold K), dataset size, and localization method.

**Strengths:**

The work is useful and practical: it reframes issue localization as a lightweight retrieve→rerank pipeline that is easy to deploy and far cheaper than agentic alternatives, while remaining compatible with standard IR/LLM components. It delivers good performance, achieving strong (often SOTA) results across file/module/function granularities on multiple benchmarks. The paper also provides insightful ablations—covering data consistency filtering, dataset scale, and hard-negative mining

**Weaknesses:**

1. Missing capacity/design ablations. The paper does not report sensitivity to model capacity or design choices for the retriever/reranker—e.g., embedding dimension, encoder depth/width. A controlled ablation study would strengthen causal claims.
2. Numerical latency study. Could you report latency for SWERank vs. agent baselines. The paper currently mentions multi-round agent latency qualitatively (Line 47, Line 125) but lacks a quantitative latency analysis for your method.
3. First-token objective may underutilize reasoning. The reranker trains only on the first generated token to select the positive, which could limit the model’s ability to use structured reasoning before deciding. A format-constrained “think-then-answer” protocol might improve generalization; loss could be defined over the formatted answer (or via RL with a verifier/reward), rather than solely first-token CE.

**Questions:**

See weakness.

---

> ### Author Response · Authors · 2025-12-02
> **Response to Reviewer ok1L (1/2)**
>
> We thank the reviewer for the constructive feedback and are glad they recognize our work as useful, practical, and effective. Below, we address the specific concerns raised.
>
> ## W1: Missing ablations on model capacity (embedding dimension, encoder depth/width)
>
> We agree that understanding sensitivity to model capacity is crucial. We note that **Table 4** in our main paper (lines 432-444) already provides experiments with both code and text pre-trained LLM **rerankers** of different sizes (3B, 7B, and 32B), demonstrating that performance generally improves as reranker size increases.
>
> To address the reviewer's request regarding **retriever** capacity and design, we conducted additional experiments. We selected the recently released Qwen3-Embedding models [1] (0.6B and 8B variants) and finetuned them on the SweLoc dataset using the same procedure as SweRankEmbed. The table below compares the recently released Qwen3-Embedding models [1] (0.6B and 8B variants) against our SweRankEmbed variants on SWE-Bench-Lite before and after finetuning:
>
> | Model | Encoder Depth (Depth) | Width (Hidden Size) | Embedding Dimension | Func. Acc@5 (Before -> After) | Func. Acc@10 (Before -> After) |
> | :--- | :--- | :--- | :--- | :--- | :--- |
> | SweRankEmbed-Small (137M) | 12 | 768 | 768 | 51.82 -> 63.14 | 58.76 -> 74.45 |
> | Qwen3-Embedding=0.6B | 28 | 2048 | 1024 | 52.55 -> 66.79 | 62.77 -> 75.18 |
> | SweRankEmbed-Large (7B) | 28 | 3072 | 3584 | 63.14 -> 71.90 | 70.44 -> 82.12 |
> | Qwen3-Embedding-8B | 36 | 3072 | 4096 | 60.95 -> **73.72** | 71.53 -> **83.94** |
>
> **Encoder Depth/Width**: Comparing SweRankEmbed-Small to Qwen3-0.6B, we see moderate gains from the significantly deeper/wider encoder. However, comparing SweRankEmbed-Large to Qwen3-8B suggests diminishing returns from further increasing depth (28 vs. 36 layers). On the other hand, the considerable gain in performance from Qwen3-0.6B to SweRankEmbed-Large seems to come from using a larger embedding dimension, which we further investigate next.
>
> **Embedding Dimension**: To isolate the impact of embedding dimension, we performed a controlled ablation using the Qwen3-Embedding-0.6B model, which supports flexible vector dimensions via Matryoshka Representation Learning (MRL) [2]. The results on SWE-Bench-Lite are as follows after finetuning Qwen3-Embedding-0.6B on SweLoc using MRL:
>
> | Embedding Dimension (D) | Func. Acc@5 (Before -> After) | Func. Acc@10 (Before -> After) |
> | :--- | :--- | :--- |
> | 1024 | 52.55 -> **66.79** | 62.77 -> **75.18** |
> | 512 | 50.00 -> 65.69 | 59.85 -> 74.09 |
> | 256 | 44.16 -> 59.12 | 52.92 -> 69.71 |
> | 128 | 39.05 -> 56.93 | 46.72 -> 66.79 |
> | 64 | 33.21 -> 50.00 | 38.32 -> 60.22 |
>
> Performance drops significantly as embedding size decreases, identifying dimension as a critical factor. Interestingly, fine tuning on SweLoc provides larger relative gains at lower dimensions (e.g., +21.9 points on Func. Acc@10 for D=64 vs. +12.5 points for D=1024), highlighting the dataset's utility even for compressed representations. We will include these analyses in the revised paper.
>
> [1] Qwen3 Embedding: Advancing Text Embedding and Reranking Through Foundation Models; Zhang et al 2025
>
> [2] Matryoshka Representation Learning; Kusupati et al NeurIPS 2022
>
> ## W2: Report latency study for SweRank vs Multiagent baselines
>
> Following the reviewer’s suggestion, we present latency numbers for our approach when compared with LocAgent.
> As the SweRankEmbed retriever has negligible inference latency (embeddings can be pre-computed), the primary cost lies in the reranker. The table below reports the average latency (over 50 instances on SWE-Bench-Lite) for our approach using **SweRankLLM-Small** (on a single 80GB A100 GPU) and GPT-4o as the reranker, compared to the **LocAgent** baseline (via GPT-4o API only as their released code doesn’t support running local models).
>
> | Approach | Model | Latency |
> | :--- | :--- | :--- |
> | SweRank | SweRankLLM-Small (A100) | **12.5s** |
> | SweRank | GPT-4o | 30.2s |
> | LocAgent | GPT-4o | 85.3s |
>
> Our approach with the local SWERANKLLM-Small reranker is approximately **7x faster** than LocAgent, while still being considerably more performant (Tables 1 and in the paper). Even when using the same model (GPT-4o), our single-turn ranking approach is nearly 3x faster than the multi-turn agentic baseline.
>
> For a more comprehensive latency study, we also analyzed the token footprint (Average Input Prompt & Output Tokens)
>
> | Approach | Prompt Tokens | Output Tokens |
> | :--- | :--- | :--- |
> | SweRankLLM | 78,409 | 741 |
> | LocAgent | 234,197 | 1884 |
>
> SweRank operates with a **~3x lower token footprint**, significantly reducing both latency and inference cost compared to the agentic baseline. We will include this analysis in the revised version.

---

> > ### Author Response · Authors · 2025-12-02
> > **Response to Reviewer ok1L (2/2)**
> >
> > ## W3: First-token objective may underutilize reasoning. A format-constrained “think-then-answer” protocol might improve generalization
> >
> > We agree that a "think-then-answer" protocol (Chain-of-Thought) could potentially unlock better reasoning for complex issues. In the early phase of this project, we had conducted a set of preliminary experiments to explore this direction using **Group Relative Policy Optimization (GRPO)**, an online reinforcement learning method. Following standard RL-for-LLM practices, we enforced an output format to separate the model’s chain-of-thought generation from the final answer and assigned a verifiable reward to the final prediction. The reward was computed as a floating-point score based on the rank position of the true label among the model’s predicted options.
> >
> > Unfortunately, we observed only minor performance gains over the standard SFT-trained counterpart. However, the inclusion of the "thinking" process introduced significant inference-time latency, undermining the primary advantage of our retrieve-and-rerank framework: its lightweight and high-throughput nature. Given the minimal performance gain versus the substantial cost in latency, we decided to maintain the efficient first-token ranking objective for this work. However, we believe this remains a promising direction for future research, particularly to push performance on hard instances.

---

### Official Review · Reviewer_NXP9 · 2025-10-25

**Soundness:** 3
**Presentation:** 4
**Contribution:** 4
**Rating:** 8
**Confidence:** 5

**Summary:**

The author proposed SWERank of an efficient method for SW issue localization. It fine-tuned two models SWERank-Embed and SWERank-LLM to retrieve and rank the issue localization in just one shot. Comparing to agent based approach, it achieves both cost and accuracy goal, as demonstrated by SWE-bench-lite and LocBench SOTA performance. The author also created a new dataset SWELoc by carefully processing public GitHub PR data, valuable resource for future research works. This work shows a purposely built non-agentic retrieval + ranking models can be more effective and practical for automated software engineering than general-purpose reasoning agents.

**Strengths:**

1. The author proposed the  retrieval + ranking models fine-tuning with open-source qwen model instead of agent using close source model, demonstrating both SOTA performance and cost efficiency. The accuracy improvement (Acc@1,3,5,10) over existing methods are significant.
2, The idea is intuitive and reasonable, the theoretical analysis is good. The novelty on top of existing work is explained well. The experiments are relatively thorough, including baselines setup, ablation study, different methods and metrics comparison.
3. The SWELoc dataset (query, positive, negatives) is a great contribution to the community.
4. Overall writing is decently good.

**Weaknesses:**

1. even though agent based approach might be costly or inefficient (i.e. not single step), it can be dynamic and up-to-date knowledge with reasoning on the results. while the embedding + LLM finetuning approach becomes static and without much reasoning support with the author's current implementation. Better tradeoff must be taken in real situation.
2. the embedding model is fine-tuned from qwen2-7b, which seems to be extremely huge for a simple embedding. This significant expense of preprocessing the index should be accounted in the cost comparison instead of retrieval and re-rank only.

**Questions:**

1. SWE is built from Python repositories (>80%). will it work as effectively for strongly typed language such as C++, Java etc.? Will it work as single model finetuning with all major programming languages?
2. do you think a hybrid approach embedding + LLM finetuning as starter to guide the multi-agent - will that combination achieve better accuracy + reasoning + up-to-date knowledge?

---

> ### Author Response · Authors · 2025-11-27
> **Response to Reviewer NXP9**
>
> We thank the reviewer for their positive assessment and for recognizing our approach as intuitive. We address the reviewer’s comments below:
>
> ## W1 & Q2: Will a hybrid approach of SweRank + Agentic system achieve better accuracy?
>
> We agree with the reviewer that integrating SweRank with agentic systems is a promising direction. Rather than viewing them as mutually exclusive, we view SweRank as a complementary, high-efficiency tool for issue localization.
>
> Current agent-based methods (e.g., SWE-Agent, LocAgent) suffer from high latency and cost because they must navigate massive search spaces using expensive LLM calls. As noted in the paper, an agent often requires 7-10 rounds of interaction. By using SweRank as a retrieval tool, an agent can instantly narrow the search space from thousands of files to a small set of high-probability candidates. This allows the agent to focus its "reasoning budget" on analyzing the retrieved code and generating the fix, rather than on navigation
>
> Therefore, a hybrid approach--using SweRank to retrieve/rerank and an agent to reason over the top-K result--is indeed the optimal real-world deployment strategy. SweRank provides the necessary speed and recall to make agentic workflows economically viable. We leave this direction for future work.
>
> ## W2: Expense of preprocessing the index with Qwen-7B embedding model could be significant
>
> We appreciate the reviewer raising the point about the cost of indexing with a 7B model. While the model size is non-trivial, the cost profile remains highly favorable for two reasons: (1) indexing is an amortized cost (performed once per repository version, not per query), and (2) even on a per-instance basis, it is negligible compared to agent inference.
>
> To demonstrate this, we calculated the indexing cost for SWE-Bench-Lite and LocBench. We utilize pricing for the Qwen3-8B-embedding family (approx. $0.01 per 1M tokens as per https://openrouter.ai/qwen/qwen3-embedding-8b) as a proxy. We calculated the total tokens per instance by summing the tokens of all functions in the repository.
>
> | Dataset (Num Instances) | Mean Tokens per Instance | Index Cost per instance | Total Benchmark Index Cost |
> | :--- | :--- | :--- | :--- |
> | SWE-Bench-Lite (274) | 2.3M | 2.3 cents | $6.30 |
> | Loc-Bench (560) | 1.5M | 1.5 cents | $8.40 |
>
> We can see that the indexing cost (2.3 cents) is minimal compared to the inference cost of a single run of the high-performing variants of the LocAgent approach (9 cents for Qwen-2.5 32B and 66 cents for Claude-3.5).
>
> ## Q1: Will SweRank work effectively for strongly typed languages such as C++, Java?
>
> Although SweLoc is constructed from Python repositories, we hypothesize that SweRank would generalize well because the underlying base models (CodeRankEmbed and GTE-Qwen2-7B-Instruct) are pretrained on massive multilingual corpora.
>
> To empirically validate this, we evaluated our models on **SWE-Bench Multilingual** [1], which includes 234 tasks across 9 languages (C, C++, Java, JavaScript, TypeScript, Rust, PHP, Ruby, and Go). We also compared our performance against Gemini-Embedding [2], a state-of-the-art proprietary general-purpose retriever trained on a variety of text and retrieval data.
>
> **Retriever Function Localization Performance on SWE-Bench Multilingual**
>
> | Model | Acc@5 | Acc@10 |
> | :--- | :--- | :--- |
> | CodeRankEmbed (137M) | 26.50 | 35.04 |
> | **SweRankEmbed-Small (137M)** | **33.33** | **44.02** |
> | GTE-Qwen2-7B-Instruct (7B) | 34.19 | 42.31 |
> | **SweRankEmbed-Large (7B)** | **39.74** | **50.85** |
> | Gemini-Embedding (unknown) | 36.75 | 47.44 |
>
> **Reranker Function Localization Performance on SWE-Bench Multilingual**
>
> | Model | Acc@5 | Acc@10 |
> | :--- | :--- | :--- |
> | SweRankEmbed-Small | 33.33 | 44.02 |
> | + CodeRankLLM | 42.74 | 51.28 |
> | + SweRankLLM-Small | **49.15** | **56.84** |
>
> **Findings**:
> - **Strong Generalization**: SWERankEmbed-Large (50.85% Acc@10) outperforms the proprietary Gemini-Embedding (47.44%), despite being finetuned on Python data.
> - **Consistent Gains**: Finetuning on SweLoc provided significant gains over the base models for both the retriever and the reranker, proving that the "issue-to-code" relevance signal learned from SweLoc transfers across programming languages.
>
> We will include these multilingual results in the revised version of the paper.
>
> [1] Swe-bench Multilingual: https://www.swebench.com/multilingual.html
>
> [2] Gemini-Embedding: Generalizable Embeddings from Gemin; Lee et al 2025.

---

### Official Review · Reviewer_8WzV · 2025-10-31

**Soundness:** 3
**Presentation:** 3
**Contribution:** 3
**Rating:** 4
**Confidence:** 4

**Summary:**

This paper presents SWERANK, a two-stage retrieve-and-rerank framework for software issue localization. The authors first use a retriever to quickly identify candidate code snippets, then apply an LLM-based reranker to refine the ranking. They also introduce a large-scale training dataset, SWELOC. Experiments on multiple benchmarks show that SWERANK outperforms existing retrieval and agent-based approaches.

**Strengths:**

- The paper is well-written and easy to follow.
- The proposed SWELOC dataset is a valuable contribution to the software engineering community.
- The authors provide thorough comparisons with multiple baselines across two benchmarks.

**Weaknesses:**

- Retrieve-and-rerank is a well-established approach, and prior work (e.g., agentless) has already applied it to software issue localization.
- The effectiveness of retrieve-and-rerank is heavily constrained by the recall of the retrieval stage; compared to agent-based methods, its upper bound may be lower.

**Questions:**

- In some cases, the bug location may have little semantic overlap with the issue description. Agent-based methods can handle this via multi-step tool-assisted navigation within the repository. How would SWERANK handle such scenarios?
- If I understand correctly, SWERANKEMBED performs retrieval at the function level. How does it handle very long functions?

---

> ### Author Response · Authors · 2025-11-27
> **Response to Reviewer 8WzV (1/2)**
>
> We thank the reviewer for their constructive feedback and for recognizing our work as a valuable contribution to the community. We address the reviewer’s concerns and questions below:
>
> ## W1: Retrieve-and-rerank is a well-established approach, and Agentless has already applied it to software issue localization
>
> ### Novelty of our approach
>
>  While we agree that the retrieve-and-rerank architecture is a well-established paradigm, our contribution lies in the **adaptation of this framework to the specific constraints of issue localization**, specifically regarding training supervision.
>
> As detailed in Section 4.2 (lines 462-479), we introduce a training strategy that allows a listwise reranker to be trained **without a full ground-truth permutation**. Unlike standard approaches (e.g., RankVicuna[1],  RankZephyr [2]) that rely on distilling expensive, closed-source models (like GPT-4) to generate a "correct" ranking of all negatives, our objective maximizes the likelihood of the single positive instance. This is specifically useful for issue localization, since software repositories possess the ground truth for the *positive* (the patch) but have no ground truth ranking for the *negative* functions. Our approach thereby enables training a state-of-the-art reranker on massive amounts of open-source data without the prohibitive cost or bottleneck of generating synthetic rankings via models like GPT-4.
>
> As shown in Table 1, this specific supervision strategy allows our open-source 32B model to outperform powerful closed-source models like GPT-4.1.
>
> ### Distinction from Agentless
>
> Agentless adopts a **LLM-based hierarchical approach** to localization instead of a retrieve-and-rerank pipeline. It first selects files based on repository structure (without file content), and then analyzes the content of those specific files to find functions. In contrast, SweRank directly performs localization at the **function-level**. Hence, our approach is considerably superior, with even the 137M parameter SweRankEmbed-Small retriever alone (58.57 Func. Acc@10 on LocBench)  outperforming the full Agentless pipeline (42.68 Func. Acc@10) using Claude-3.5, as seen in Table 2 in the paper.
>
> [1] RankVicuna: Zero-Shot Listwise Document Reranking with Open-Source Large Language Models; Ronal et al 2023.
>
> [2] RankZephyr: Effective and Robust Zero-Shot Listwise Reranking is a Breeze!, Ronak et al 2024
>
> ## W2: Effectiveness of retrieve-and-rerank is heavily constrained by the recall of the retrieval stage; compared to agent-based methods, its upper bound may be lower.
>
> We agree that the reranker is theoretically bounded by the retriever's recall (Acc@100). To address the reviewer’s concern regarding the "ceiling" provided by the retriever, we provide extended retrieval metrics below (Function Acc@20, @50, @100) for both benchmarks. We also include Gemini-Embedding [3], a proprietary SOTA retrieval model (Top-4 on MTEB).
>
> **SWE-Bench-Lite Function Retrieval Performance**
>
> | Model | Acc@10 | Acc@20 | Acc@50 | Acc@100 |
> | :--- | :---: | :---: | :---: | :---: |
> | SweRankEmbed-Small | 74.45 | 81.75 | 87.96 | 91.97 |
> | SweRankEmbed-Large | 82.12 | 86.50 | 90.88 | **93.43** |
> | Gemini-Embedding | 72.26 | 79.20 | 87.96 | 90.88 |
>
> **LocBenchFunction Retrieval Performance**
>
> | Model | Acc@10 | Acc@20 | Acc@50 | Acc@100 |
> | :--- | :---: | :---: | :---: | :---: |
> | SweRankEmbed-Small | 58.57 | 67.50 | 75.71 | 82.32 |
> | SweRankEmbed-Large | 63.21 | 71.25 | 80.71 | **84.29** |
> | Gemini-Embedding | 51.43 | 60.18 | 70.00 | 78.39 |
>
> On SWE-Bench-Lite, our large retriever achieves **93.4% Acc@100**. Our final reranker achieves **88.7% Acc@10**. The gap suggests that the retrieval stage is not yet the primary bottleneck, and our retrieval ceiling (93.4%) is significantly higher than the best performance achieved by agentic methods like LocAgent (Acc@10 ~78%). Likewise for LocBench. We will include these results in the revised version of the paper.
>
> [3] Gemini-Embedding: Generalizable Embeddings from Gemini; Lee et al 2025.
>
> ## Q2: How does SweRankEmbed handle long functions?
>
> We thank the reviewer for raising this point. We performed a statistical analysis (shown below) of function-level token counts (as per the Qwen-2.5 tokenizer) across the repositories in our benchmarks.
>
> | Dataset | Mean | Median | 99th Percentile |
> | :--- | :--- | :--- | :--- |
> | SWE-Bench-Lite | 230 | 103 | 1340 |
> | LocBench | 263 | 132 | 1961 |
>
> We observe that the vast majority of functions are short (mean < 300 tokens), with the 99th percentile being roughly 2000 tokens. For functions that are considerably longer, which represent a very minor fraction of all the functions, we simply truncate to the first 2048 tokens. We note that our base models (Arctic and Qwen) support longer contexts (8k and 32k). We chose the 2048 limit primarily for GPU memory efficiency during batch encoding, but this can be easily scaled up if a specific repository contains exceptionally large functions.

---

> > ### Author Response · Authors · 2025-11-27
> > **Response to Reviewer 8WzV (2/2)**
> >
> > ## Q1: How does SweRank handle bug localization with little overlap to issue description?
> >
> > We thank the reviewer for raising this interesting question.To address this, we analyzed performance by grouping instances based on **Lexical Overlap** (Rouge-1) and **Semantic Overlap** (Cosine Similarity with Gemini-Embedding).
> >
> > ### Lexical Overlap (Rouge-1)
> >
> > For lexical overlap, we bucket instances into 4 groups using the Rouge-1 score between the issue description and ground-truth localized functions for that instance. A high rouge-score indicates high lexical overlap. For instances with multiple positives, we take the mean of the individual Rouge-1 scores. The tables below show performance for different approaches, with the relative distribution of the buckets also provided. We show numbers separately for the SweRankEmbed retriever variants, along with the SweRankLLM-large reranker run using the SweRankEmbed-Large retriever outputs. (We show numbers for LocAgent only on SWE-Bench-Lite since the authors do not release model outputs for LocBench)
> >
> > **SWE-Bench-Lite Performance (Function Acc@10) Breakdown by Rouge score**
> >
> > | Model | Rouge-1: 0 - 0.1 (16%) | Rouge-1: 0.1 - 0.15 (24%) | Rouge-1: 0.15 - 0.2 (23%) | Rouge-1: 0.2-0.4 (37%) |
> > | :--- | :---: | :---: | :---: | :---: |
> > | LocAgent (Claude-3.5) | 60.87 | 64.62 | 74.60 | 81.00 |
> > | Gemini-Embedding | 58.70 | 72.31 | 69.84 | 80.00 |
> > | SweRankEmbed-Small | 63.04 | 66.15 | 74.60 | 85.00 |
> > | SweRankEmbed-Large | 65.22 | 84.62 | 80.95 | 89.00 |
> > | +SweRankLLM-Large | 78.26 | 87.69 | 87.30 | 95.00 |
> >
> > **Loc-BenchPerformance (Function Acc@10) Breakdown by Rouge score**
> >
> > | Model | Rouge-1: 0 - 0.1 (25%) | Rouge-1: 0.1 - 0.15 (26%) | Rouge-1: 0.15 - 0.2 (22%) | Rouge-1: 0.2 - 0.4 (27%) |
> > | :--- | :---: | :---: | :---: | :---: |
> > | Gemini-Embedding | 34.75 | 40.14 | 59.20 | 72.11 |
> > | SweRankEmbed-Small | 42.55 | 47.62 | 67.20 | 77.55 |
> > | SweRankEmbed-Large | 47.52 | 53.74 | 68.00 | 83.67 |
> > | +SweRankLLM-Large | 57.45 | 65.99 | 75.20 | 86.39 |
> >
> > Based on the results, we observe that the performance of the models gradually decreases as lexical overlap decreases.
> > Even in the lowest overlap bucket (0.0 - 0.1), SweRankEmbed-Large outperforms LocAgent, demonstrating that our model does not rely solely on keyword matching and captures semantic relationships effectively. The SweRankEmbed-Large retriever further improves performance, with significant gains seen specifically for instances with low lexical overlap.
> >
> > ### Semantic Overlap (Cosine Similarity)
> >
> > For semantic overlap, we bucket instances into 4 groups based on the mean cosine similarity between the embeddings of the issue description and ground-truth localized functions. The embeddings are computed based on the general-purpose Gemini-Embedding model. A high cosine similarity implies greater semantic overlap. The tables below show performance for different approaches when instances are grouped by cosine similarity.
> >
> > **SWE-Bench-Lite Performance (Function Acc@10) Breakdown by Cosine Similarity**
> >
> > | Model | Cos. Sim:0.65-0.75 (19%) | Cos. Sim: 0.75 - 0.775 (31%) | Cos. Sim: 0.775 - 0.8 (25%) | Cos. Sim: 0.8 - 0.9 (25%) |
> > | :--- | :--- | :--- | :--- | :--- |
> > | LocAgent (Claude-3.5) | 52.94 | 72.62 | 86.96 | 94.12 |
> > | Gemini-Embedding | 16.98 | 67.86 | 92.75 | 100.00 |
> > | SweRankEmbed-Small | 41.51 | 71.43 | 79.71 | 98.53 |
> > | SweRankEmbed-Large | 49.06 | 82.14 | 92.75 | 97.06 |
> > | +SweRankLLM-Large | 58.49 | 92.86 | 95.65 | 100.00 |
> >
> > **Loc-Bench Performance (Function Acc@10) Breakdown by Cosine Similarity**
> >
> > | Model | Cos. Sim:0.65-0.75 (32%) | Cos. Sim: 0.75 - 0.775 (24%) | Cos. Sim: 0.775 - 0.8 (24%) | Cos. Sim: 0.8 - 0.9 (20%) |
> > | :--- | :--- | :--- | :--- | :--- |
> > | Gemini-Embedding | 10.11 | 36.96 | 81.48 | 100.00 |
> > | SweRankEmbed-Small | 29.78 | 47.83 | 76.30 | 97.25 |
> > | SweRankEmbed-Large | 32.58 | 53.62 | 85.19 | 98.17 |
> > | +SweRankLLM-Large | 41.01 | 65.94 | 93.33 | 100.00 |
> >
> > The results show that, as expected, the performance of retriever models is directly correlated to semantic overlap, with near-perfect performance for instances with high cosine similarity (>0.8). The SweRankEmbed-large reranker considerably improves performance over the retriever, specifically for instances with low semantic overlap, while outperforming the multi-turn LocAgent approach across all the buckets.
> >
> > This indicates that while agentic tool-use is one way to bridge the semantic gap, our training on SweLoc (which includes hard negatives) allows the retriever/reranker to learn these non-obvious mappings effectively. We will include these results in the revised version.

---

### Official Review · Reviewer_mEZB · 2025-11-01

**Soundness:** 3
**Presentation:** 3
**Contribution:** 2
**Rating:** 4
**Confidence:** 4

**Summary:**

This paper presents SWERANK, a retrieve-and-rerank framework for software issue localization, i.e., identifying code locations related to natural language issue descriptions such as bug reports. Experiments on SWE-Bench-Lite and LocBench show that SWERANK achieves state-of-the-art performance, surpassing agent-based systems (e.g., LocAgent with Claude-3.5) at all localization granularities.
It also offers significant cost advantages, at only $0.011–$0.015 per instance, compared to $0.46–$0.66 for agent-based approaches.

**Strengths:**

- The paper is well-structured.

- Practical significance and efficiency. The paper targets the underexplored yet practically important problem of software issue localization. SWERANK offers a simple, efficient alternative to costly multi-step LLM agents. Its cost-effectiveness (up to 40–60× cheaper than Claude-based agents) enhances its industrial applicability.

**Weaknesses:**

- **Limited methodological novelty**. The overall framework (retrieve-then-rerank) and component designs largely reuse existing paradigms. SWERANKEMBED is a standard bi-encoder trained with InfoNCE loss, similar to prior works such as CodeRankEmbed (2022–2024). SWERANKLLM simplifies listwise reranking to predicting the positive sample ID, which resembles earlier weakly supervised rankers like RankVicuna.
Given the rapid development of retrieval and reranking methods by 2025, the paper’s contribution is mainly empirical rather than conceptual. The authors should better justify why this design suits issue localization uniquely, beyond cost reduction.

- **Lack of independent component analysis**.
The two modules are evaluated only in combination. For SWERANKEMBED, there is no analysis of standalone retrieval quality (e.g., recall@N). If recall is low (e.g., <60% at Top-20), the reranker’s potential impact becomes limited. For SWERANKLLM, the authors do not report results when reranking the full candidate set independently. Demonstrating that each module performs reasonably well on its own would strengthen the technical soundness of the retrieve-then-rerank pipeline.

- **Insufficient analysis on issue complexity** The differences between Acc@5, Acc@10, and Acc@15 are small. It remains unclear whether issue complexity (e.g., single-function vs. cross-module dependencies) influences localization difficulty.
Grouping issues by dependency complexity or code change scope could reveal where the reranker struggles, providing deeper insight into model limitations.

**Questions:**

Q1: Could the authors provide independent evaluations of SWERANKEMBED and SWERANKLLM?
For example, what is the standalone retrieval recall of SWERANKEMBED at different Top-N settings, and how well does SWERANKLLM rerank full candidate lists without relying on the retriever’s output?

Q2: Given that both modules adopt well-established designs (bi-encoder retrieval with InfoNCE and listwise reranking), how does SWERANK differ conceptually from prior retrieve-and-rerank systems like CodeRankEmbed or RankVicuna?
What makes its design particularly suited to issue localization beyond efficiency gains?

---

> ### Author Response · Authors · 2025-11-27
> **Response to Reviewer mEZB (1/2)**
>
> We thank the reviewer for their constructive feedback and for recognizing the practical significance of our work. Below, we clarify the reviewer’s concerns about our methodological contributions and provide deeper analysis regarding component performance and issue complexity:
>
> ## W1 & Q2: Methodological novelty and how the design is specific to Issue Localization
>
> While we agree that the retrieve-and-rerank architecture is a well-established paradigm, our contribution lies in the **adaptation of this framework to the specific constraints of issue localization**, specifically regarding training supervision.
>
> As detailed in Section 4.2 (lines 462-479), we introduce a training strategy that allows a listwise reranker to be trained **without a full ground-truth permutation**. Unlike standard approaches (e.g., RankVicuna[1],  RankZephyr [2]) that rely on distilling expensive, closed-source models (like GPT-4) to generate a "correct" ranking of all negatives, our objective maximizes the likelihood of the single positive instance. This is specifically useful for issue localization, since software repositories contain the ground truth for the *positive* (the patch) but have no ground truth ranking for the *negative* functions. Our approach thereby enables training a state-of-the-art reranker on massive amounts of open-source data without the prohibitive cost or bottleneck of generating synthetic rankings via models like GPT-4.
>
> As shown in Table 1, this specific supervision strategy allows our open-source 32B model to outperform powerful closed-source models like GPT-4.1.
>
> [1] RankVicuna: Zero-Shot Listwise Document Reranking with Open-Source Large Language Models; Pradeep et al 2023.
>
> [2] RankZephyr: Effective and Robust Zero-Shot Listwise Reranking is a Breeze!; Pradeep et al 2024
>
> ## W2 & Q1: Evaluation of SweRankEmbed and SweRankLLM Components separately
>
> ### Standalone Retriever Performance
>
> We respectfully point out that the standalone performance of SweRankEmbed reported in the “Retriever” rows of Tables 1 and 2. In our experimental setup (Section 5.1), the metric **Acc@K** is a strict functional equivalent to Recall@K, as it considers a localization successful *only if all* modified functions are present in the Top-K results.
>
> To address the reviewer’s concern regarding the "ceiling" provided by the retriever, we provide extended retrieval metrics below (Function Acc@20, @50, @100) for both benchmarks. We also include Gemini-Embedding [3], a proprietary SOTA retrieval model (Top-4 on MTEB).
>
> **SWE-Bench-Lite Function Retrieval Performance**
>
> | Model | Acc@10 | Acc@20 | Acc@50 | Acc@100 |
> | :--- | :---: | :---: | :---: | :---: |
> | SweRankEmbed-Small | 74.45 | 81.75 | 87.96 | 91.97 |
> | SweRankEmbed-Large | 82.12 | 86.50 | 90.88 | **93.43** |
> | Gemini-Embedding | 72.26 | 79.20 | 87.96 | 90.88 |
>
>
> **LocBench-Lite Function Retrieval Performance**
>
> | Model | Acc@10 | Acc@20 | Acc@50 | Acc@100 |
> | :--- | :---: | :---: | :---: | :---: |
> | SweRankEmbed-Small | 58.57 | 67.50 | 75.71 | 82.32 |
> | SweRankEmbed-Large | 63.21 | 71.25 | 80.71 | **84.29** |
> | Gemini-Embedding | 51.43 | 60.18 | 70.00 | 78.39 |
>
> On SWE-Bench-Lite, our large retriever achieves **93.4% Acc@100**. Our final reranker achieves **88.7% Acc@10**. The gap suggests that the retrieval stage is not yet the primary bottleneck, and our retrieval ceiling (93.4%) is significantly higher than the best performance achieved by agentic methods like LocAgent (Acc@10 ~78%). Likewise for LocBench. We will include these results in the updated version of the paper.
>
> ### Feasibility of Full Candidate-Set Reranking
>
> Regarding the evaluation of SweRankLLM on the full candidate set,  we clarify that independent reranking of the entire repository is computationally infeasible. Repositories in SWE-Bench-Lite contain an average of **9.9k functions** (90th percentile: 27k), while those in LocBench contain an average of **5.5k functions** (90th percentile: 10.1k). Running a 32B or even 7B parameter LLM on ~10k candidates per issue would reintroduce the high latency and cost our framework is designed to eliminate. The retrieve-then-rerank design is therefore a structural necessity for software scale, rather than just a design choice.
>
> [3] Gemini-Embedding: Generalizable Embeddings from Gemini; Lee et al 2025.

---

> > ### Author Response · Authors · 2025-11-27
> > **Response to Reviewer mEZB (2/2)**
> >
> > ## W3: Analysis of Issue Complexity, e.g. single function vs cross-module dependencies.
> >
> > We thank the reviewer for raising this interesting point, and agree that aggregate metrics can obscure performance variance on complex issues.  Following the reviewer's suggestion, we stratified the test set by **Num_Gold** (the number of functions modified in the ground truth patch) as a proxy for issue complexity. We compare our approach against LocAgent (SOTA Agentic) and Gemini-Embedding [3] (SOTA proprietary embedding). (We show numbers for LocAgent only on SWE-Bench-Lite since the authors do not release model outputs for LocBench)
> >
> > ### SWE-Bench-Lite Performance (Function Acc@10) Breakdown by Complexity:
> >
> > | Approach | Num_Gold = 1 (87% of data) | Num_Gold = 2-3 (13% of data) |
> > | :--- | :--- | :--- |
> > | LocAgent (Claude 3.5) | 82.01 | 47.06 |
> > | Gemini-Embedding | 75.21 | 54.29 |
> > | SweRankEmbed-Small | 77.73 | 54.29 |
> > | SweRankEmbed-Large | 82.77 | 77.14 |
> > | +SweRankLLM-Large | **89.92** | **80.00** |
> >
> > ### LocBench Performance (Function Acc@10) Breakdown by Complexity:
> >
> > | Approach | Num_Gold = 1 (63%) | Num_Gold = 2-3 (22%) | Num_Gold = 4-5 (8%) | Num_Gold > 5 (7%) |
> > | :--- | :--- | :--- | :--- | :--- |
> > | Gemini-Embedding | 69.32 | 29.51 | 17.78 | 0.0 |
> > | SweRankEmbed-Small | 75.57 | 42.62 | 20.00 | 2.44 |
> > | SweRankEmbed-Large | 78.12 | 50.00 | 35.56 | 4.88 |
> > | +SweRankLLM-Large | **85.51** | **62.30** | **40.00** | **9.76** |
> >
> > **Observations**:
> > - As expected, performance degrades as the number of modified functions increases. Instances requiring changes to >5 functions are extremely difficult for all models (single-digit accuracy).
> > - SweRankEmbed-Large demonstrates significantly better scaling on complex issues (Num_Gold =2-3) compared to Agentic approaches (LocAgent drops from 82% to 47%, while our retriever maintains 77%).
> > - The reranker consistently improves performance across all complexity levels, confirming that it successfully captures cross-function dependencies that the bi-encoder might miss.

---

### Meta-Review · Area_Chair_W978 · 2026-01-06

**Summary:**

This paper addresses the task of identifying specific code locations relevant to natural language issue descriptions like bug reports. While recent LLM-based agentic approaches show promise, they have significant latency and cost due to multi-step reasoning . The authors propose SWERANK, an efficient retrieve-and-rerank framework that formulates issue localization as a single-shot ranking problem rather than an agentic reasoning task. A key methodological contribution is the training strategy for the reranker, they modify the listwise objective to only predict the positive function's identifier rather than a full permutation, enabling training on massive open-source data without expensive synthetic ranking generation. The authors also construct a large-scale dataset curated from 3,387 public GitHub repositories, featuring real-world issue descriptions paired with corresponding code modifications, processed through consistency filtering and hard-negative mining for quality. Experiments on SWE-Bench-Lite and LocBench demonstrate that SWERANK achieves state-of-the-art performance across file, module, and function-level localization, outperforming both prior ranking models and costly agent-based systems while being much cheaper.

**Reviewer Concerns:**

Reviewers raised some major concerns:
(1) limited methodological novelty, as retrieve-and-rerank is well-established and Agentless already applied it to issue localization.
(2) lack of independent component evaluation, particularly standalone retrieval recall metrics.
(3) concern that retriever recall bounds overall performance, potentially giving agent-based methods a higher ceiling.
(4) missing ablation studies on model capacity and issue complexity.

The authors provided comprehensive responses. They clarified that their key contribution is a training strategy enabling listwise rerankers without full ground-truth permutations, avoiding expensive distillation. Extended retrieval metrics  showed the retriever ceiling significantly exceeds agentic methods. Complexity analysis demonstrated better scaling on multi-function issues.

While the retrieve-and-rerank paradigm is established, the authors demonstrate meaningful adaptation for issue localization. The empirical results are strong and comprehensive, with SOTA performance at dramatically reduced cost. The SWELOC dataset is a valuable resource for the community. The rebuttal thoroughly addressed reviewer concerns with extensive new experiments on retrieval ceiling, model capacity, latency, issue complexity, and multilingual generalization.

**Reviewer Scores:**

mEZB: 4->6: All concerns comprehensively addressed with new experiments and analysis.
8WzV: 4)->4/6: Retriever ceiling and semantic overlap analyses directly address main concerns.
NXP9: 8->8:  Already positive; multilingual experiments strengthen the paper.
ok1L: 6->6:  Capacity ablations and latency study address the main weaknesses.

---

### Decision · Program_Chairs · 2026-01-26

Accept (Poster)